# Application of embedded soft PLC in the control system of rapier loom

Yanjun Xiao[1,2]☯*, Linhan Shi [1,2]☯, Wei Zhou[1]☯, Feng Wan[1]☯, Weiling Liu[1]☯*

1 School of Mechanical Engineering, Tianjin Key Laboratory of Power Transmission and Safety Technology for New Energy Vehicles, Hebei University of Technology, Tianjin, China, 2 Career Leader Intelligent Control Automation Company, Suqian, Jiangsu Province, China

☯ These authors contributed equally to this work.
* 13622021167@163.com (YX); a13622021167@163.com (WL)

**Data Availability Statement:** All relevant data are within the manuscript and its Supporting information files.

**Funding:** This work was supported by Jiangsu Province training fund funded project (Grant No.

## Abstract

At present, the rapier loom has gradually become the mainstream equipment in the manufacturing industry. In order to make the rapier loom realize automated production and further improve the production efficiency of the rapier loom, improve the programmability of the system, and reduce the cost of system maintenance. The thesis developed a rapier loom control system based on embedded soft PLC, and carried out experiments and applications in the field. The contribution and innovation of this paper is to develop a complete low-cost control system, and through a genetic algorithm optimized PID algorithm to complete the more effective control of the loom tension system. The embedded soft PLC system proposed in this paper reduces the overall maintenance cost of the system and improves the programmability of the system. This text carries on the systematic scheme design to the embedded soft PLC from the hardware system and the software system respectively. First, according to the actual requirements, this article designs the overall scheme of the embedded software PLC hardware system with STM32F407ZGT6 as the core. Then this article is based on the embedded soft PLC hardware platform, according to the international standard of industrial control programming, writes the embedded soft PLC low-level driver software. Secondly, this article analyzes the factors that affect the warp tension during the operation of the rapier loom, and proposes the use of genetic algorithm to optimize the warp tension control method of the traditional PID algorithm. Finally, we conducted verification tests and on-site application debugging for the entire set of rapier loom embedded soft PLC control system. We controlled the warp tension as the main experimental object. The results show that this control system effectively improves the control accuracy of the warp tension of the rapier loom and meets the actual needs of industrial applications. The whole system has a good application prospect in the warp tension control of rapier looms.

## 1. Introduction

Now, users of rapier looms not only require higher reliability and stability of the loom, but also put forward higher expectations for future looms [1,2]. At present, there is still a big gap

BRA2020244). This funded project is used to support the research on the intelligent control system of the rapier loom. This project is presided over by Xiao Yanjun. The Career Leader Intelligent Control Automation Company provided an experimental platform for this project. Career Leader intelligent control automation company provided support in the form of salaries for author Yanjun Xiao, but did not have any additional role in the study design, data collection and analysis, decision to publish, or preparation of the manuscript. The specific roles of these authors are articulated in the 'Author contributions' section.

**Competing interests:** The author Yanjun Xiao and Linhan Shi is employed by Career Leader intelligent control automation company. This does not alter our adherence to PLOS ONE policies on sharing data and materials. The authors have declared that no competing interests exist.

between China's loom level and foreign developed countries. Major foreign rapier loom manufacturers have made a lot of improvements and efforts in realizing high-speed, intelligent, automated, multi-applicability, and modularization of looms [3–6]. For example, the GS920 rapier loom launched by the Italian SMIT company is a new type of rapier loom based on the "SMART" public platform. The PICANOL company of Belgium has recently launched a new generation of OPTIMAX rapier looms, adopting a modular design concept. Each loom is based on a unified platform for easy upgrade and improvement. Among the main products recently launched by the German company DORNIER, the PTS series rigid rapier loom adopts a unique positive central weft transfer method, which can produce a variety of difficult industrial fabrics [7–10]. In summary, foreign rapier looms have fully applied the latest and comprehensive modular design concepts to expand and evolve the original model.

Most foreign rapier loom manufacturers use single-chip microcomputers as the control core, so that they are at a relatively high level in terms of operating speed, stability and upgrade speed. However, the maintenance cost of the operation and control system of these looms is very high, and the system programming method of the single-chip control core is not simple, and it is not suitable for users to quickly get started and use.

From the analysis of the production status of domestic textile machines, most manufacturers use PLC as the control core and equipped with a rotary photoelectric encoder as the detection original. This design not only simplifies the hardware design, reduces the point of failure, but also improves the reliability and stability of the system, and its programmability features greatly reduce maintenance costs, which has been affirmed by users.

Although the maintenance cost is greatly reduced, because the foreign PLC production technology is far higher than the domestic level, the market has basically been monopolized by foreign manufacturers, so the cost of purchasing these systems is very high. Some manufacturers will be discouraged by the high cost when choosing the corresponding PLC as the core of their control system.

These phenomena indicate that the electrical control part of the current rapier loom is mainly embodied in two ways: single-chip control and PLC control. Both methods have their own advantages and disadvantages. From the perspective of the single-chip control system, the overall production cost of the single-chip control system is relatively low, and it is easy to develop from research and development to manufacturing. The shortcomings of the single-chip control system are poor programmability, and the quality of electronic components cannot be guaranteed during the research and development process, which leads to higher maintenance costs for the control system with the single-chip microcomputer as the core purchased by enterprises in large batches. Compared with the PLC control system, the single-chip control system has the biggest advantage of the system with PLC as the control core is high reliability and stability, and strong anti-interference ability. Compared with the shortcomings of the high maintenance cost of the single-chip microcomputer, the PLC system has programmability and operability. Most of the functions in the PLC control system are realized by software technology, which can reduce the use of peripheral hardware, thereby reducing the maintenance cost of the system. Considering the advantages of PLC control system's programmability, low maintenance cost, and later upgrade and development, more and more manufacturers choose PLC as the control core [11,12]. This shows that once these two technologies are combined, the control system equipment of the rapier loom will not only have the characteristics of low single-chip cost and easy production, but also have the characteristics of simple PLC programmability and low maintenance cost. The result of the combination of technologies will surely be able to occupy a leading position in the market.

After comparing existing research in related directions, it is found that hierarchical or monitoring control based on embedded soft PLC is usually used in industrial processes. Some

recent work reports on industrial information systems indicate that there is currently no configurable platform for designing such control systems. The excavator control system based on embedded soft PLC proposed by Xu Xiao realizes switch and analog processing, logic control and general communication interface through the software resources of the system, and gets rid of the situation that the core module of the dedicated controller is restricted by people. Although this design has the advantages of strong data processing ability, open system, strong network communication ability and low cost, it is still not satisfactory in terms of system programmability [13]. Gao Yanxiang designed an automatic control system using embedded soft PLC technology in view of the high research and development costs of the comprehensive excavation automatic control system currently used with ordinary PLC supporting special controllers, large amount of maintenance and inconsistent research and development platforms. The system can complete the basic logic control, remote control and automatic cutting functions of the comprehensive excavator. However, there are still great difficulties in system programming and development [14]. In view of the low control success rate of existing intelligent controllers in the market, excessively long control process time, and poor overall performance, Shi Chunxiao designed a new intelligent controller based on embedded soft PLC technology. The controller makes the control success rate higher and the consumption time is shorter. However, there is not much mention in the later maintenance cost of the system and the programmability of the system [15]. Zhu Wei and others designed a control system based on embedded soft PLC technology to solve the problems of high development cost, large maintenance, and difficulty in cross-platform transplantation in the existing coal mine excavator control system with ordinary PLC and special controller as the core. Compared with the control system with PLC and special controller as the core, the excavator control system using embedded soft PLC can better realize the standard unification of excavator equipment and system configuration development. However, the entire system focuses on the system's cross-platform portability, and the system's later maintenance cost and system programmability still fail to meet the existing requirements [16]. DAI et al. proposed a configurable OSC platform called OS Control. The goal is to enable relevant personnel to independently work on control systems without software development skills. However, the development of this kind of information system monitoring controller is a difficult and error-prone process, and it cannot effectively improve the programmability of the system [17]. Aiming at the design and testing of some industrial process monitoring systems, Zhou et al. developed a process monitoring system platform based on HILS. The design of the monitoring system is realized through the development of monitoring controllers and loop controllers based on embedded soft PLC, but it cannot effectively reduce system maintenance costs [18]. In order to optimize energy consumption, Celestine selects the most suitable Cluster Head (CH) in the IoT network. A hybrid meta-heuristic algorithm is proposed, namely simulated annealing algorithm (SA) and whale optimization algorithm (WOA) [19]. In this paper, Praveen designed a hybrid whale optimization algorithm-Moth Flame Optimization (MFO) to select the optimal CH [20]. This gives this article an inspiration to merge two different control systems into an independent embedded soft PLC system to achieve the advantages and goals that a single single-chip control system and a single PLC control system cannot achieve. The success of the debugging of the embedded soft PLC system in this paper also proves the feasibility of the embedded soft PLC control system proposed under the guidance of this fusion idea in solving the related problems of the rapier loom control system.

In summary, in some industrial control such as monitoring control, the single-chip control system and the PLC control system have their own advantages. The research goal of this paper is to integrate the advantages of the single-chip control system and the PLC control system, reduce the maintenance cost of the system and improve the programmability of the system,

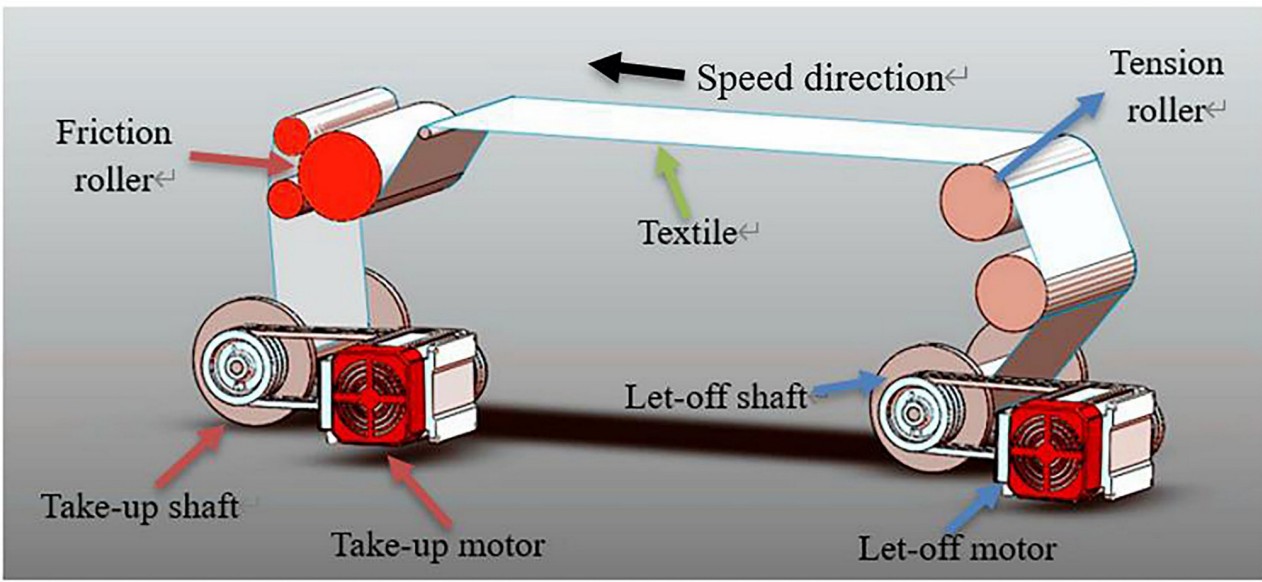

**Fig 1. Structure drawing of rapier loom model.**

thereby further improving the production efficiency of the rapier loom. This subject proposes and independently develops an embedded soft PLC control system. The contribution and innovation of this article lies in the development of a complete low-cost control system, that is, an embedded soft PLC control system. In this paper, through the design of the hardware structure and software structure of the embedded soft PLC control system, at the same time design the embedded soft PLC bottom drive system and the embedded soft PLC control system main program. The use of STM32 single-chip microcomputer as the main control unit and PLC programming method effectively reduces the maintenance cost of the system and effectively improves the programmability of the system. Compared with other control methods in the field of rapier looms, this system solves the problems of high maintenance cost and poor programmability that have occurred in the control system of rapier looms in the past. And the control system proposed in this article has a wide range of applications, which is suitable for rapier loom control systems generally used in actual industrial production. After that, based on this system platform, we completed a more effective control of the tension system of the loom through a PID algorithm optimized by genetic algorithm. From the actual test results on site, it is confirmed that the system can operate stably for a long time in the on-site environment, reducing the system maintenance cost and improving the system programmability, and at the same time improving the production efficiency of the rapier loom. The structure drawing of rapier loom model is shown in Fig 1.

## 2. Research goals, content and innovations

### 2.1 Research goals of the rapier loom control system based on embedded soft PLC

Aiming at the on-site rapier loom, design a set of feasible embedded soft PLC control system, focusing on the design and realization of the hardware circuit system and the design and realization of the soft PLC bottom drive system and experimental verification.

In the hardware system construction, a 32-bit microprocessor can be used as the core, with power-down protection storage. At the same time, the EMC characteristics are improved by using electrical isolation and other methods, and the electrical isolation design parameter tuning method is proposed, so that the Ethernet and bus expansion functions can meet the application requirements of Industry 4.0. Responding to the functional requirements of the embedded soft PLC control system, the whole system includes hardware modules such as power supply module, minimum CPU system, communication interface, on-chip bus, digital input and output, analog input and output, and dedicated control blocks.

The software structure part includes embedded soft PLC bottom layer software and application software. Embedded soft PLC bottom layer software adopts special programming software to design the soft PLC drive system. In order to realize the basic definition of the embedded soft PLC framework, the function of each module is correspondingly designed. The underlying drive system of the soft PLC can compile the PLC program, so that the embedded control platform can recognize the PLC ladder diagram program and realize the application of the PLC ladder diagram program [21,22]. Finally, according to the process requirements of the controlled object rapier loom, an application program for controlling the rapier loom is designed. Real-time monitoring of the production line is realized by designing the corresponding human-computer interaction interface.

The experimental verification part is based on the embedded soft PLC system control platform designed in this paper. The result of controlling the tension system of the loom through a PID algorithm optimized by genetic algorithm shows the superiority of the control system designed in this paper in improving the actual production efficiency of the loom. At the same time, the experimental results also show the application value of the embedded soft PLC control system designed in this paper with low maintenance cost and programmability in actual industrial production.

## 2.2 The innovation of the rapier loom control system based on embedded soft PLC

1. For process and system integration issues, the hardware platform of the embedded soft PLC control system is designed according to specific process requirements, which changes the high maintenance cost of the embedded control platform of rapier looms in the past.

2. Through the process analysis and research, the corresponding application software is designed to realize the communication between the upper computer software platform and the embedded PLC platform, which solves the shortcomings of poor programmability of the embedded control system of the rapier loom.

3. By analyzing the factors that affect the warp tension during the operation of the rapier loom, a genetic algorithm is proposed to optimize the warp tension control method of the traditional PID algorithm. The results of simulation through MATLAB software and final debugging on the actual industrial site show that the warp tension stability of the rapier loom is improved, the number of warp breaks is reduced, and the weaving efficiency of the loom is significantly improved. The Schematic diagram of rapier loom fabric formation is shown in Fig 2.

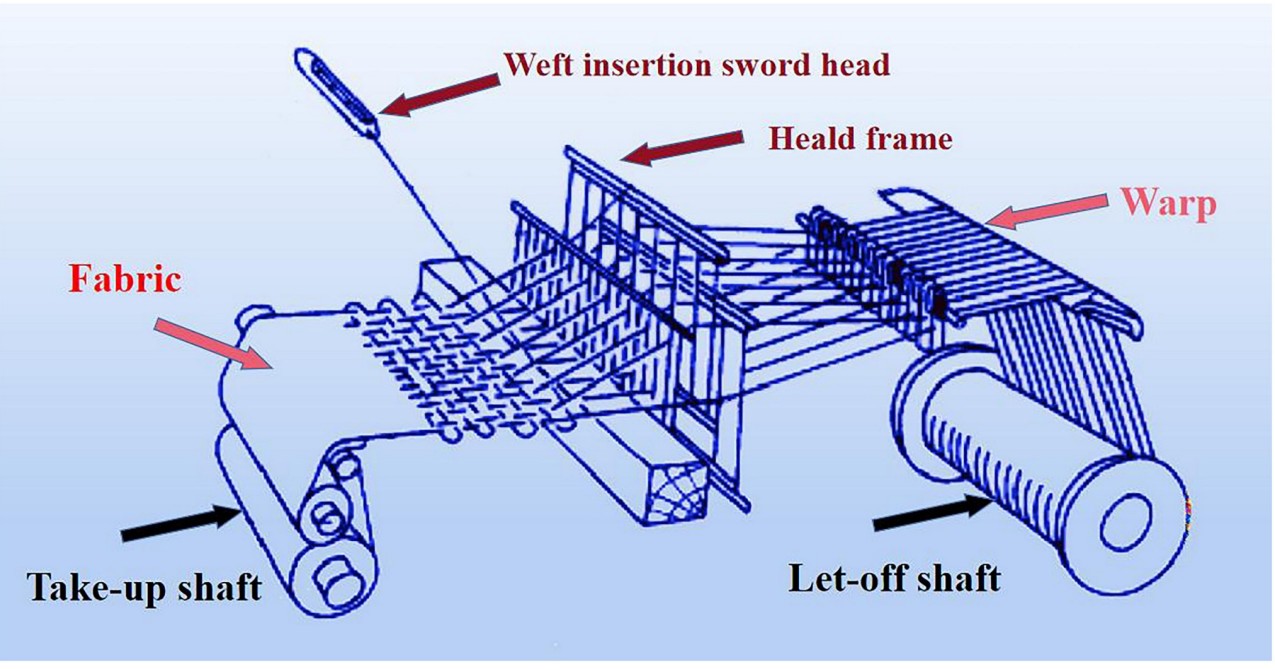

**Fig 2. Schematic diagram of rapier loom fabric formation.**

## 3 The structure of the embedded soft plc control system of the rapier loom

### 3.1 The overall hardware structure of the embedded soft PLC control system of the rapier loom

Traditional PLC is a combination of relay and computer science and technology. Its basic structure is composed of memory, central processing unit, power supply, input and output unit, input and output expansion interface, external device interface and programmer module. The CPU is the control center of the PLC. The CPU stores the user's data and application programs while interpreting the PLC program, and then checks the status of various peripherals and ports and various errors in the writing process. PLC memory is generally divided into system memory and user memory, which are used to store system programs and user programs, respectively. The input and output unit is the connection port between the PLC and the input and output equipment of the industrial field. The external device interface can realize the communication with the touch screen and other devices [23].

According to the hardware structure of the traditional PLC, this topic uses STM32F407ZGT6 as the control core of the embedded software PLC hardware platform to expand the system's Flash memory. The external input/output unit and other peripheral units are designed in accordance with the technological requirements of the rapier loom, and the multi-power conversion unit provides power for the entire system. The hardware platform structure diagram of the embedded soft PLC control system is shown in Fig 3.

### 3.2 The software overall structure of the embedded soft PLC control system of the rapier loom

The software structure of the embedded soft PLC control system of this subject is divided into two parts. The first part is to use Keil uVision5 software to develop and design the STM32F407

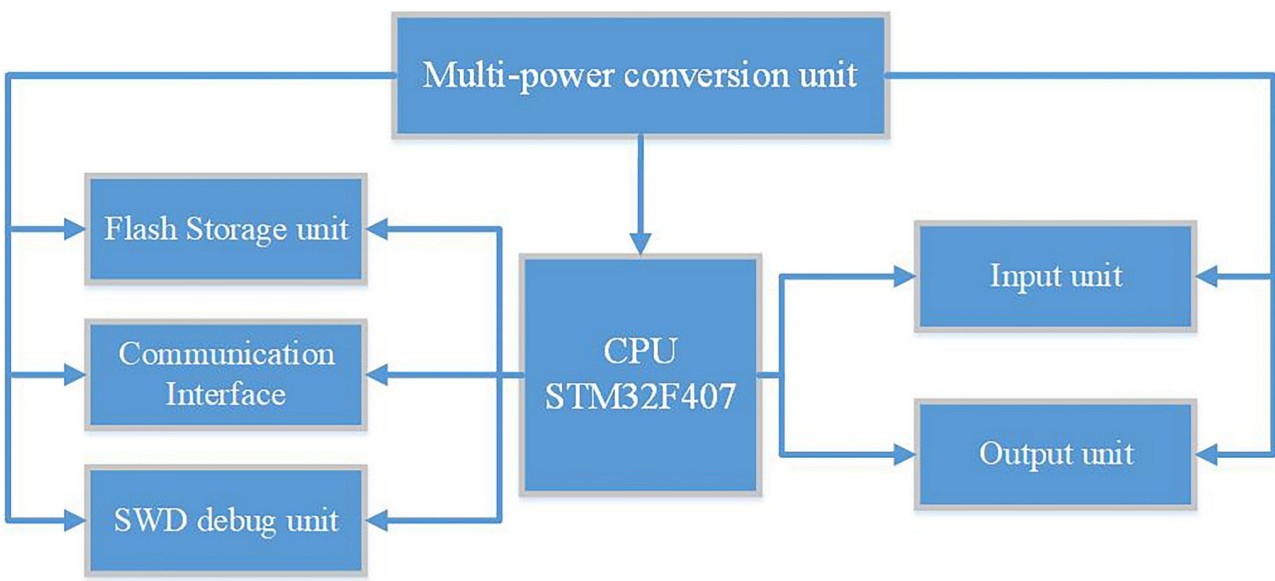

**Fig 3. Hardware platform structure of embedded soft PLC control system.**

operating system program. The program configures the overall architecture of the embedded soft PLC, and the input and output parts of the system are continuously scanned until the user program code is received and executed. The second part is to use the PLC upper computer programming software to write the program. By using GX Developer software or GX Works2 software to program, save, modify the ladder diagram and related instruction set and other basic functional attributes, complete the basic operation of the embedded PLC instruction set and realize the functions of programming from the ladder diagram of the embedded soft PLC to the instruction list. The software overall structure diagram of the embedded soft PLC control system of the rapier loom is shown in Fig 4.

## 4 Software structure design of embedded soft PLC control system

### 4.1 The design of the underlying software structure of the embedded soft PLC control system

**4.1.1 The working principle of traditional PLC.** Traditional PLC works in a certain order. A scan cycle of PLC is composed of input sampling, program execution and output refresh, and its working principle diagram is shown in Fig 5. In a PLC ladder diagram program, the PLC will execute sequentially according to the corresponding relationship between the programs until the program runs. After the end, the PLC will return to the first instruction and restart scanning [24].

**4.1.2 Working principle of embedded soft PLC system.** The working principle of the embedded soft PLC control system proposed in this paper is similar to that of the traditional PLC. The main process is to compile the ladder diagram operation program through the corresponding software on the computer and download it to the embedded soft PLC control system. Embedded soft PLC also adopts the working mode of continuous loop and continuous scanning. The embedded soft PLC first receives the input signal of the input interface and transfers it to the input status register. Then by sequentially executing the judgment and operation of the instruction, the corresponding output data is generated. Finally, the running result is output through the output port of the embedded soft PLC control system, so as to realize the

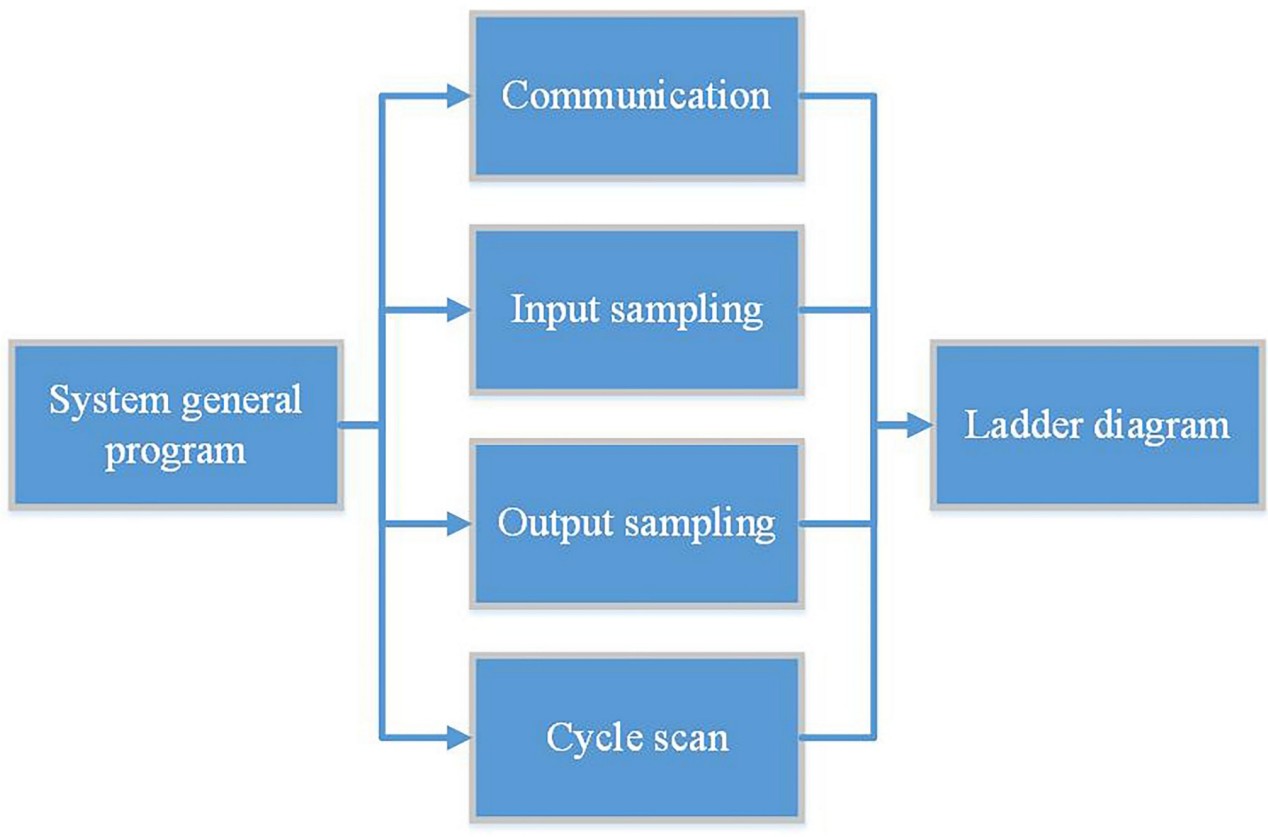

**Fig 4. Software structure diagram of embedded soft PLC control system for rapier loom.**

control of the external equipment [25]. The overall design sketch of application software is shown in Fig 6.

## 4.2 Bottom software driver design of embedded soft PLC control system

The driver module in the embedded soft PLC proposed in this article is the focus of the software part. The design idea of the drive module in the embedded soft PLC control system is to divide the whole system into various modules for programming, such as initialization module,

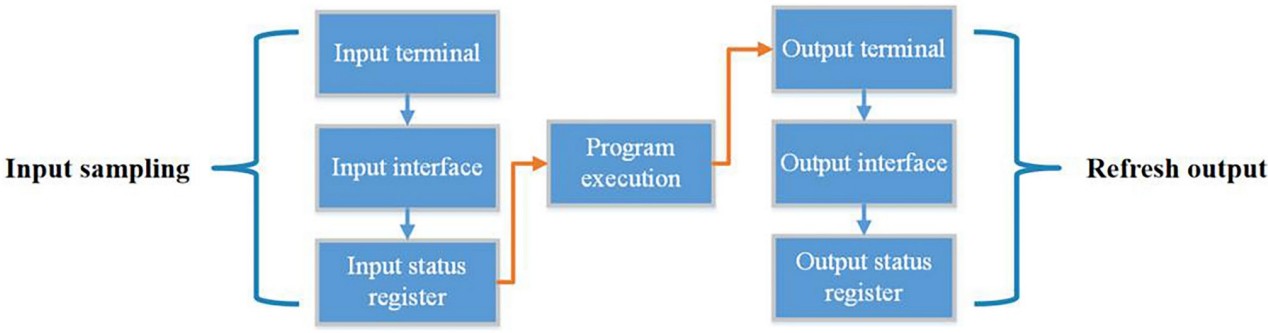

**Fig 5. PLC working principle operation diagram.**

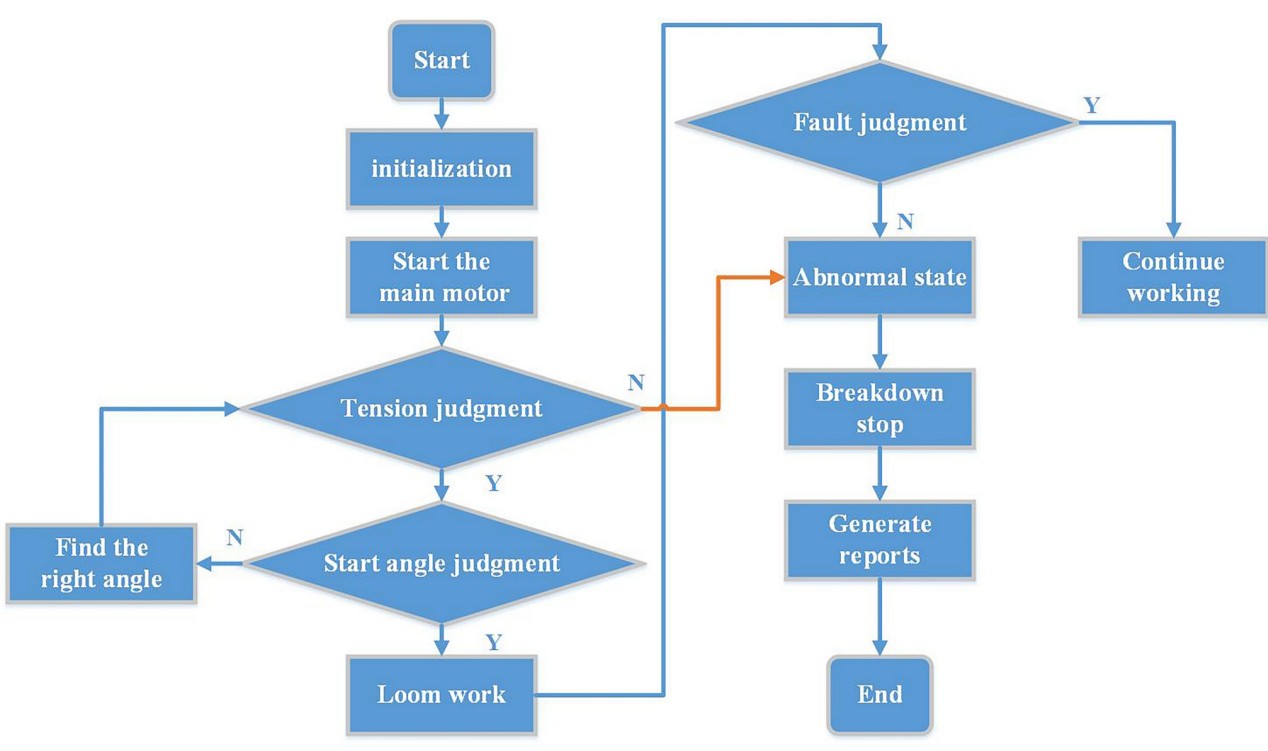

**Fig 6. Overall design sketch of application software.**

input and output module, serial communication part and other modules. Firstly, these modules are compiled separately, and then integrated and placed in the main function. Finally, the cycle scanning function of the embedded PLC is used to realize the function of each design module of the main function, and the structure of the driving function of the embedded soft PLC control system can be realized.

**4.2.1 The principle of the underlying software ladder diagram of the embedded soft PLC control system.** Interpretative PLC mainly includes the upper computer development system and the lower computer operating system. The host computer development system is to use the corresponding software on the PC to realize the functions of writing ladder diagram programs, simulation, and conversion of instruction statements. The lower computer operating system is the embedded software PLC hardware platform mentioned above, which is used to solidify the PLC analysis program.

The interpretative PLC lower computer operating system first initializes the operating system, and then takes the intermediate code corresponding to the ladder diagram edited in the host computer development system as input, and performs logic check, grammar check, byte checksum calculation, etc. Secondly, scan the input terminal, store the input information in the input image area, then scan the intermediate code, interpret and execute the intermediate code to generate the corresponding logic result, and store the logic result in the output image register. Finally, scan the output terminal and use the output result in the output image register to drive the external device. The schematic diagram of the overall structure of the interpretative PLC is shown in Fig 7.

The following takes a program as an example to introduce the analysis process of PLC ladder diagram: The Schematic diagram of PLC ladder program is shown in Fig 8.

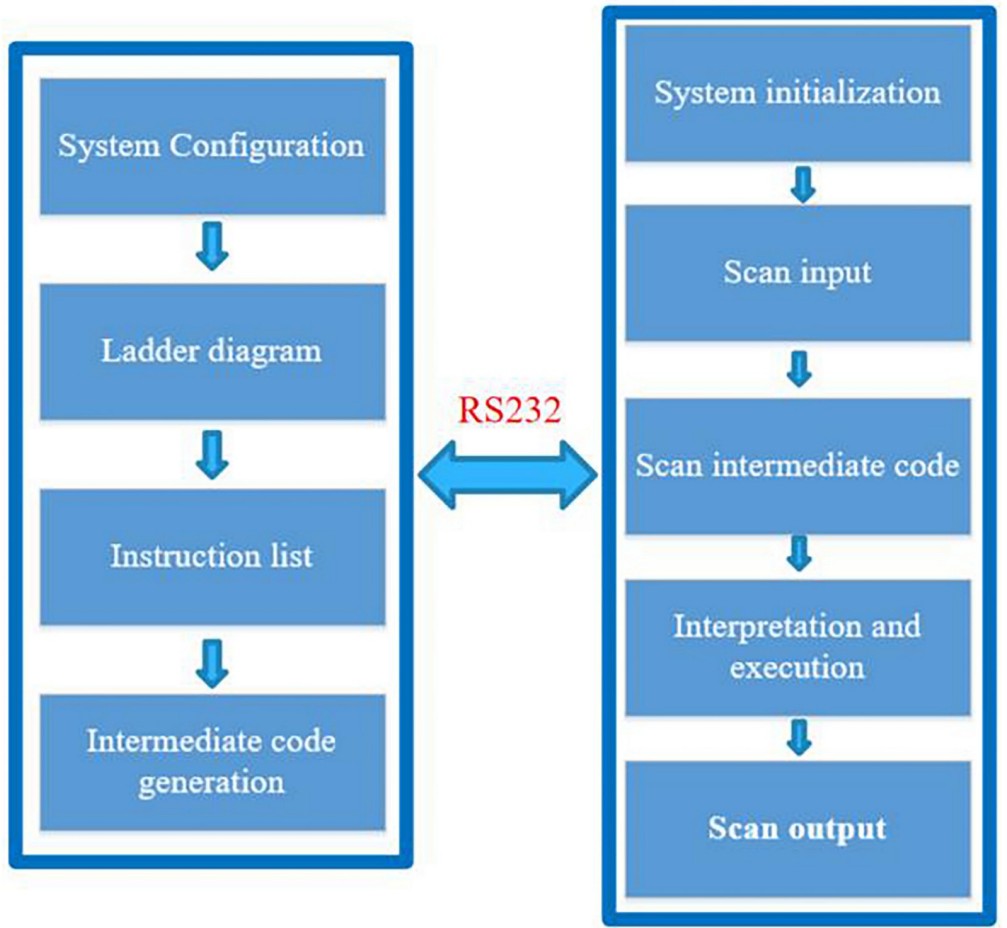

**Fig 7. Interpretation type PLC overall structure.**

The corresponding instruction list is:

$$
\begin{aligned}
&LD \quad X0\\
&OR \quad Y0\\
&LDI \quad X1\\
&OUT \quad Y0
\end{aligned}
$$

END

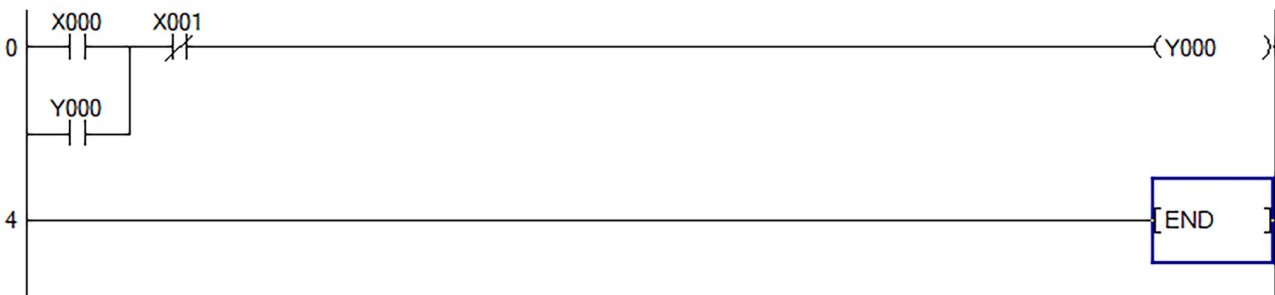

**Fig 8. Schematic diagram of PLC ladder program.**

Express it as Boolean algebra and the result is:

$$Y0 = (X0 + Y0) * \overline{X1} \tag{1}$$

It can be concluded that the subsequent expression calculation method is: First, an empty operand stack is established. If the operand is fetched, it will be stored in the stack. If the operand is fetched, the corresponding operand will be taken out of the stack to perform the corresponding operation, and the result of the operation will be stored on the stack. Finally, after all operations are completed, the stack content is the operation result of the subsequent expression.

In order to facilitate understanding, we set the corresponding soft components X0, X1, Y0 in formula (1) to "1" when they are on, and "0" when they are not. Therefore, the analysis principle of the soft PLC can be analyzed. When the analysis program of the running system encounters the LD instruction, the state of the soft element X0 is stored in the stack. The PLC continues to scan to the OR instruction, and then performs an OR operation between the current scan result and the result previously stored in the stack, and stores the result on the top of the stack. At this time, the stack stores a new data. Continue to scan and encounter the LDI instruction, and the scanned X1 data will be AND with the previous stack data, and the result will be stored on the top of the stack. Continue to scan until the END instruction is scanned, and the scan ends, and output the state "0" or "1" at the top of the stack at this time to obtain the final result.

**4.2.2 Command analysis of embedded soft PLC control system.** For the embedded soft PLC control system of the rapier loom designed in this article, after loading a PLC ladder program, the embedded soft PLC system's analysis of each instruction in this PLC ladder program is the key to this control system to realize the soft PLC function.

After the embedded soft PLC loads a program, it first reads the input port to see if any signal is read in the input port, then enters the PLC application program and reads the first instruction through the pointer. After that, point the pointer to the next instruction and continue down to determine whether the read instruction is an END instruction. If it is the END instruction, the soft PLC system will end, enter the write output port directly, and begin to read the input port cyclically. Then execute the process of PLC instruction and write port. If the END instruction is not read at this time, the operand of the line is judged. If it is X, assign the state of the corresponding X to itemp, and point to the PLC instruction of that line. If there is no X soft element, judge whether there is soft element Y in the operand of the row, If there is soft element Y, assign the state of soft element Y to itemp, and point to the PLC instruction of this line. The PLC instruction analysis program flow chart is shown in Fig 9.

The soft PLC system executes sequentially downwards, and sequentially judges and compiles the auxiliary relays, timers, counters, data registers and other soft elements. Finally, when all the soft components are judged, enter the PLC instruction judgment of this line. If the judgment is successful, the corresponding operation is performed and the next instruction pointed to by the pointer is continued. If the corresponding instruction is not judged, the soft PLC system will also point the pointer to the next instruction, and finally loop and execute the above process in turn.

## 4.3 The main program structure design of embedded soft PLC control system

The main program of the embedded soft PLC control system adopts a sequential control method around the technological requirements of the rapier loom, and the software program is designed in the form of organization blocks. First, the system will enter the initialization

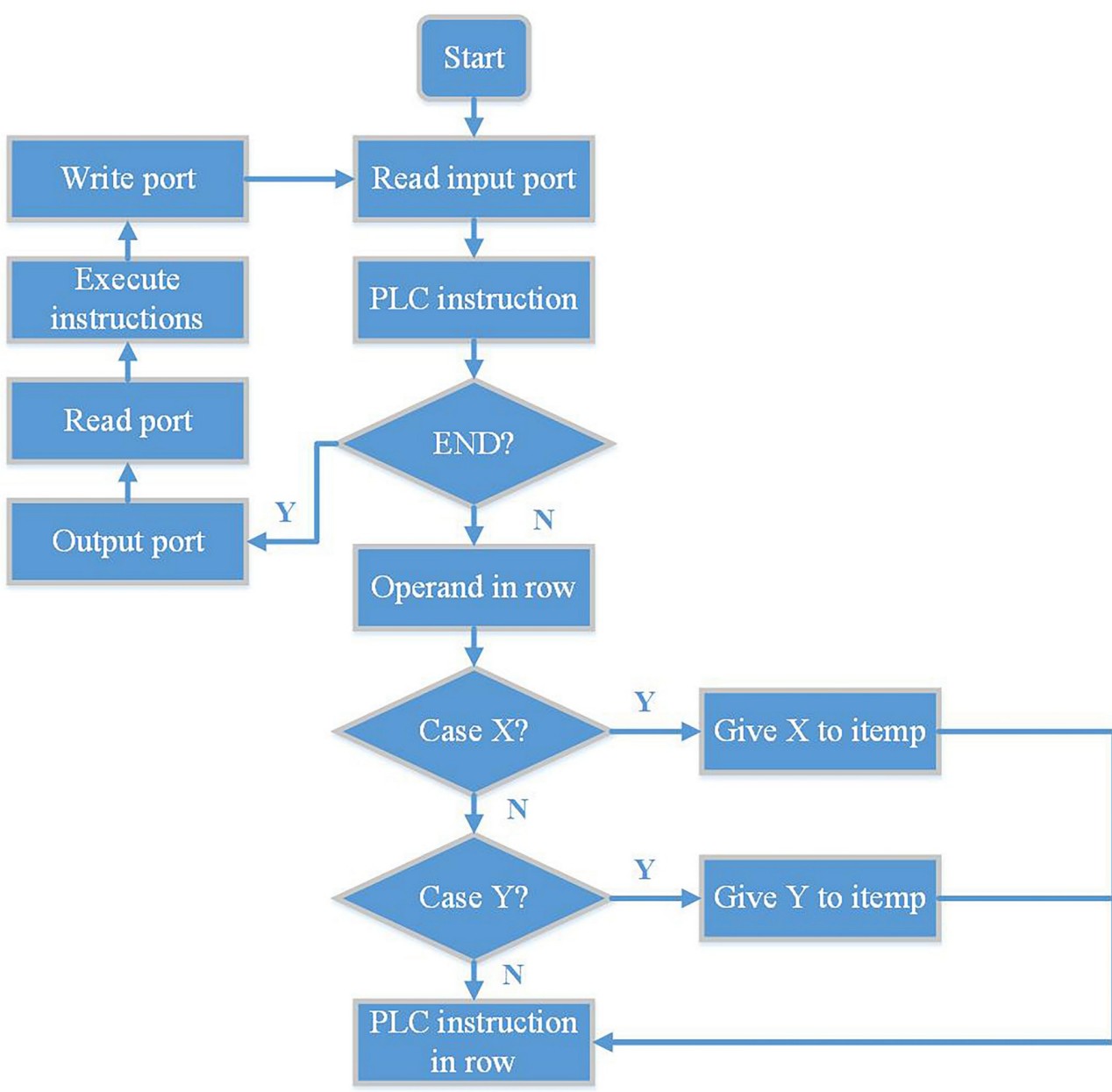

**Fig 9. PLC instruction analysis program flow chart.**

process, the initialization process is mainly centered around the needs of the rapier loom process control. Secondly, after the initialization is completed, the system first starts the main motor, and the rotation of the main motor supplies oil to the system. At this time, it detects whether the signal of the oil pressure sensor is normal. If it is normal, continue to run down. If it is abnormal, wait for the main motor to rotate and continue to supply oil until the oil pressure is normal. After the detection system is normal, there will be a process of judging the spindle angle. If it is at the starting angle, the system directly enters the weaving process. If it is not at the starting angle, the system will first search for the starting angle by the slow motor, and then continue to run downwards until the starting angle is found. During the normal

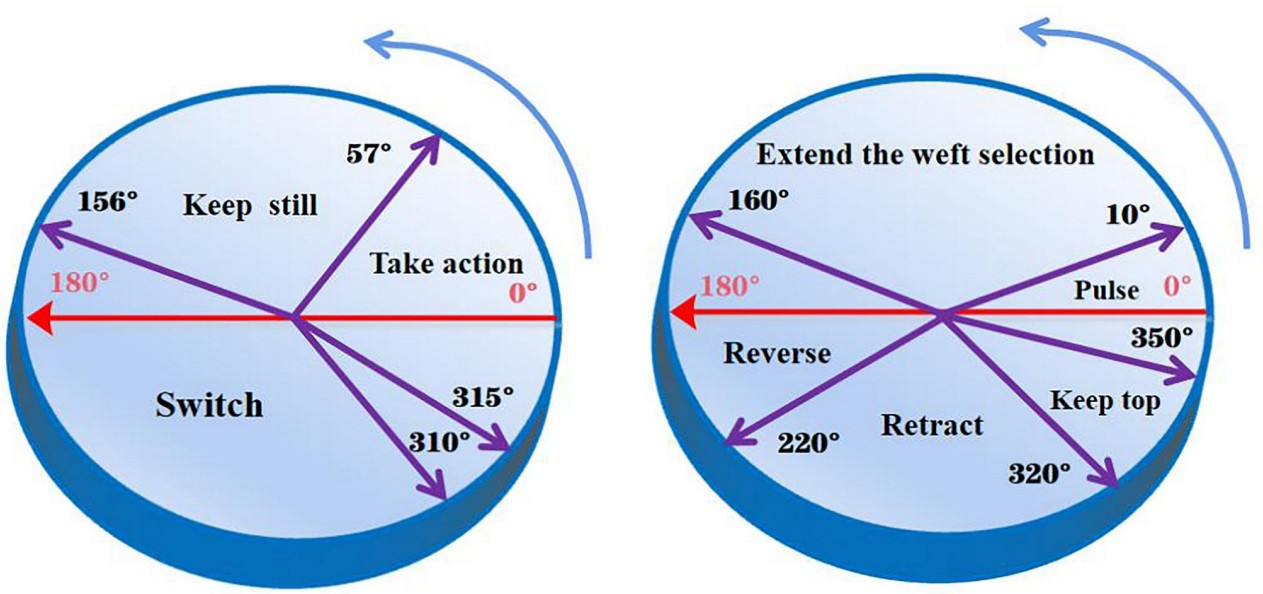

**Fig 10. The corresponding action timing chart of the spindle rotation angle.**

operation of the system, it will always detect the occurrence of failures and deal with them accordingly. The corresponding action timing chart of the spindle rotation angle is shown in Fig 10.

The acquisition and calculation process of the angle signal is as follows:

1. Since the signal obtained by the absolute encoder is Gray code, it is necessary to convert the Gray code to binary and store the binary number in the D0 register.

$$C_n = R_n$$
$$C_{n-1} = R_n \oplus R_{n-1}$$
$$C_{n-2} = R_n \oplus R_{n-1} \oplus R_{n-2} \qquad (2)$$
$$\ldots\ldots$$
$$C_0 = R_n \oplus R_{n-1} \oplus \ldots\ldots \oplus R_1 \oplus R$$

2. Turn the binary code to the angle value, and get the angle according to the following formula:

$$\text{angle} = \text{D0} * \frac{2^8}{360} \qquad (3)$$

## 5 PID parameter tuning optimization based on genetic algorithm

### 5.1 The application of traditional PID algorithm in tension control system

The warp tension system of the rapier loom is a system with real-time changes, nonlinearity and many interference factors. However, the tension of the rapier loom not only needs to be controlled in the let-off mechanism and the take-up mechanism, but also needs to be well controlled in each mechanism of the entire rapier loom. Because the tension of the rapier loom is

a key factor for the normal weaving of the rapier loom. Therefore, the control of the tension system of the rapier loom is the key to the application of the soft PLC control system proposed in this topic.

In the current industrial control, PID control algorithm has the characteristics of mature technology, simple principle and good control effect. Many fields have achieved good results by using PID algorithm. Therefore, with the rapid development of industrial technology, PID algorithm can still maintain a fresh vitality in the field of industrial control.

The key of PID algorithm control is to determine the three control parameter values of proportional, integral and derivative. In the PID controller, three parameters have a decisive influence on the control performance of the PID algorithm. The main function of the introduction of the proportional parameter is to make the controller immediately produce a control function proportional to the error signal e(t), but an excessively large ratio will cause the system to be unstable, even if it is used alone, it may cause steady-state errors. The integral parameter can play a role in eliminating the steady-state error of the system, but the integral parameter will cause the system response speed to slow down. Therefore, if the integral parameter is too large, it will reduce the stability of the system. Differential parameters can produce advanced control effects, thereby speeding up the response speed of the system, and the differential parameters can reflect the changing trend of the deviation signal. It can be seen that the prerequisite for the application of the PID algorithm is to have a correct introduction of the specific parameter values of the proportional constant, the integral time constant and the derivative time constant. However, the warp tension system of the rapier loom is a system with real-time changes, non-linearity and many interference factors. Therefore, if the PID algorithm is directly used to control the loom, it is very difficult to set specific PID parameter values. The structure diagram of PID control algorithm is shown in Fig 11.

## 5.2 PID parameter tuning optimization based on genetic algorithm

The genetic algorithm was first proposed by Professor J. Holland in the United States. It is a method to obtain the optimal solution to a practical problem by simulating the process of biological evolution and using the principles of genetics. Genetic algorithm can be used to

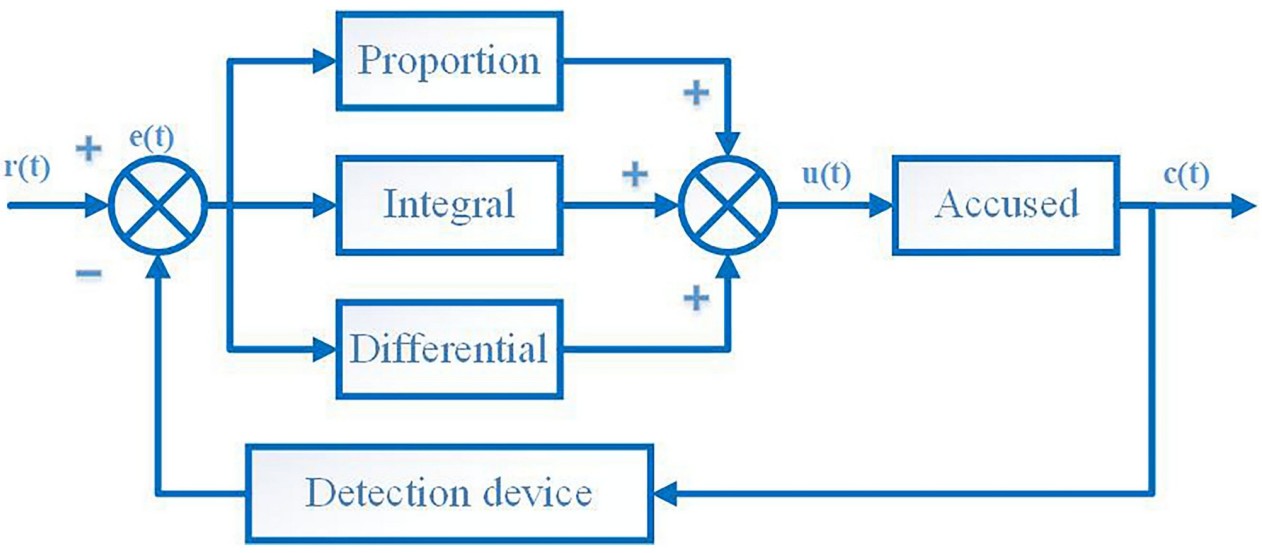

**Fig 11. Structure diagram of PID control algorithm.**

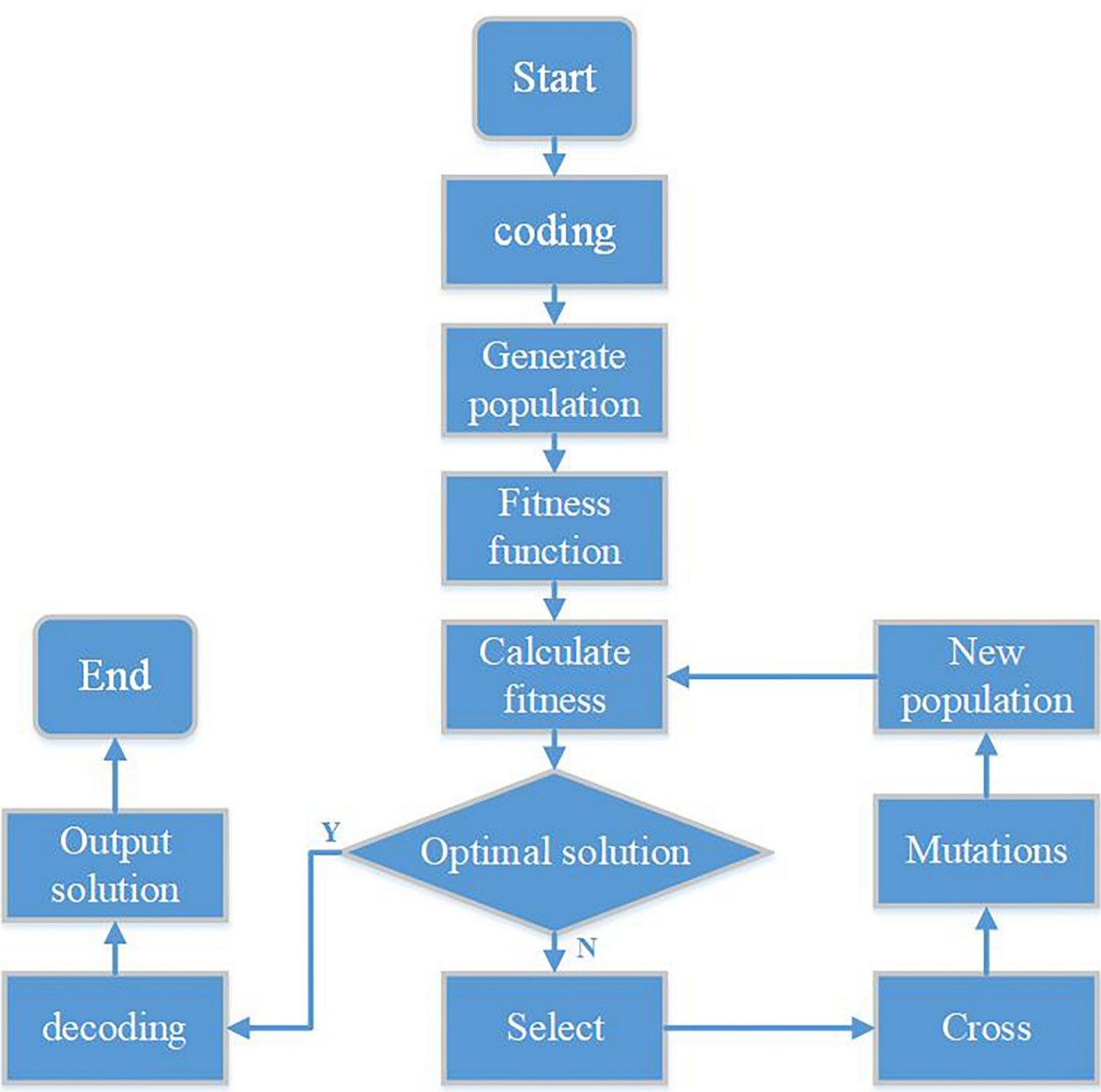

**Fig 12. Flow chart of genetic algorithms.**

optimize the solution of global problems, and its essence is a search method. Genetic algorithm is similar to natural evolution. It solves problems by acting on genes on chromosomes to find the best chromosomes. Similar to nature, genetic algorithm knows nothing about the problem itself, all it needs is more reproduction opportunities for the chromosomes produced by genetic algorithm. In the genetic algorithm, a number of digital codes, namely chromosomes, of the problem to be solved are randomly generated, and the initial population is formed; After that, a numerical evaluation is given to each individual through the fitness function, and then individuals with low fitness are eliminated, and individuals with high fitness are selected to participate in genetic operations. Finally, the individuals after genetic manipulation are assembled into a new population of the next generation, and this new population undergoes the next round of evolution [26,27]. The flow chart of genetic algorithms is shown in Fig 12.

The warp tension system of the rapier loom is a system with real-time changes, non-linearity and many interference factors. Therefore, when the PID algorithm is directly used to adjust the warp tension, the three parameters of the accurate proportional coefficient, integral coefficient and differential coefficient cannot be effectively determined. For the optimization of PID algorithm, the three parameters of proportional coefficient, integral coefficient and differential coefficient are mainly optimized and set [28–32].

The main steps proposed in this paper to optimize PID parameters through genetic algorithm are as follows:

1. Determination of initial population parameters;
   First, it is necessary to determine the value range of the PID parameters. In order to narrow the search range, this article uses the critical ratio method to tune the PID parameters. Then, set the integral coefficient and the differential coefficient to the maximum and minimum. At this time, the controller only uses the pure proportional $K_p$ function for regulation. In the control process, the value of $K_p$ is continuously adjusted, and the value of the critical proportional coefficient $\lambda_k$ and the value of the critical period $T_k$ at the moment is recorded when the constant amplitude oscillation output is debugged, and the setting value of each parameter is obtained by calculation. If there is no constant amplitude oscillation output, the maximum $K_p$ value is taken as the critical proportional coefficient $T_k$. In this paper, the value range of the proportional coefficient $K_p$ [0,20], the value range of the integral coefficient $T_i$ [0,1], and the value range of the differential coefficient $T_d$ [0,1] are obtained through testing.

2. Find the corresponding fitness function;
   The determination of the fitness function is the key to the optimal solution. Optimal tuning of the three PID parameters of $K_p$, $T_i$ and $T_d$ is the fitness function achieved through algorithms. The fitness function can be determined according to the three conditions of accuracy, rapidity and stability. In this paper, the time error integral is used as the system performance index. In order to prevent the energy from being too large, the square term of the control input is added to the objective function. In order to get a better control effect, the rise time condition is added to the constraint condition.

3. Deterministic genetic operator
   The genetic algorithm determines each group of PID parameters by selecting methods to leave high-quality individuals. In order to ensure the diversity of samples, new individuals need to be constantly added during the selection process. This paper adopts an adaptive adjustment method when determining the crossover probability and mutation probability. The function of the adaptive algorithm is to set the constant value of the crossover probability and the mutation probability in the traditional genetic algorithm into a way that can be adaptively changed according to the individual fitness situation. The advantage of this approach is that it can achieve a certain balance between the randomness of selection and the speed of convergence. Through calculation, the final formulas for crossover probability ($P_c$) and mutation probability ($P_m$) are as follows:

$$P_c = \begin{cases} k_1 \dfrac{f_{\max} - f^*}{f_{\max} - f_{avg}}, f^* \geq f_{avg} \\ k_3 f^* \leq f_{avg} \end{cases} \tag{4}$$

$$P_{\mathrm{m}} = \begin{cases} k_2 \dfrac{f_{\max} - f^*}{f_{\max} - f_{avg}}, f^* \geq f_{avg} \\[2ex] k_4 f^* \leq f_{avg} \end{cases} \tag{5}$$

The principle block diagram of PID control algorithm based on genetic algorithm is shown in Fig 13.

### 5.3 Simulation of genetic algorithm PID control

This paper uses Matlab software to build PID algorithm based on genetic algorithm to verify the optimization effect of genetic algorithm on the three parameters of PID [27,28]. The existing warp tension model is as follows [33]:

$$G(S) = \frac{0.4516\left(\dfrac{1}{7.4619}s + 1\right)}{\left(\dfrac{1}{10.2267}s + 1\right)\left(\dfrac{1}{0.037}s + 1\right)} \tag{6}$$

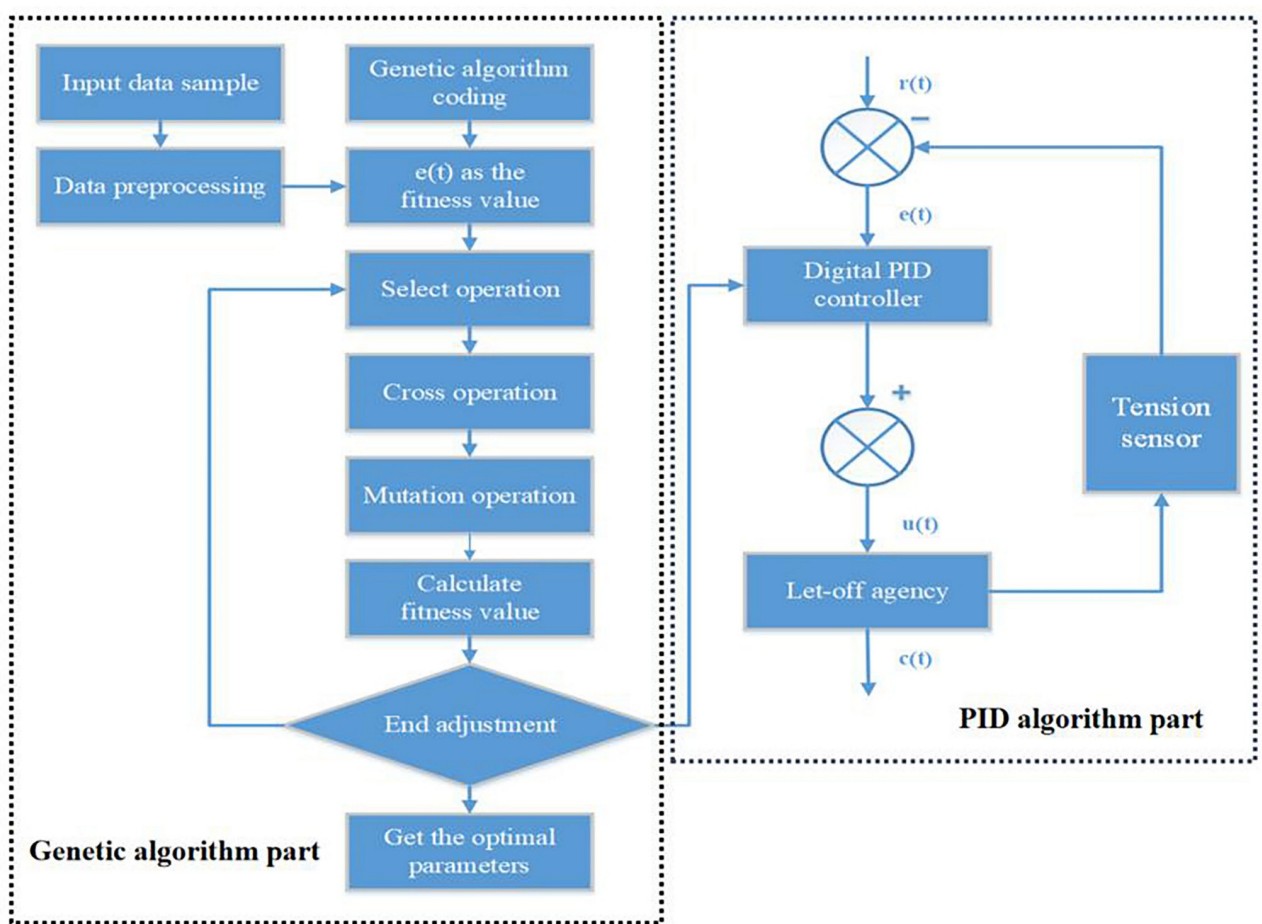

**Fig 13. Principle block diagram of PID control algorithm based on genetic algorithm.**

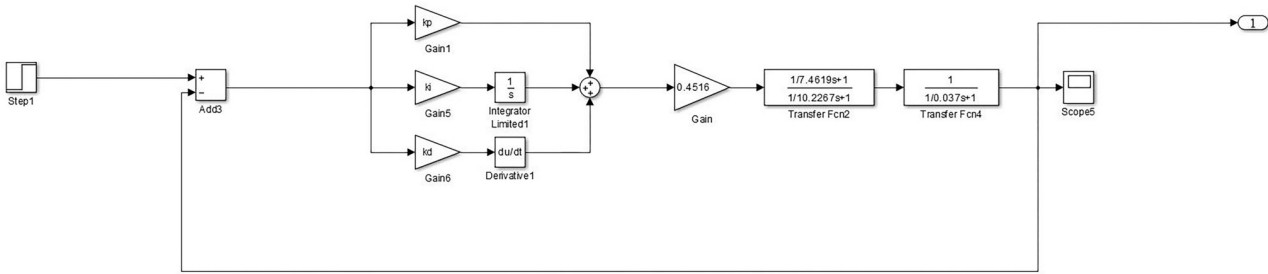

**Fig 14. Simulink simulation of genetic PID control for warp tension.**

The formula (6) warp tension model is built through the Simulink in Matlab to build a genetic PID algorithm simulation model. As shown in Fig 14.

Edit the genetic algorithm related program through Matlab, and call the model built in Fig 14 for calculation. The range of the parameter $K_p$ is [0,20], the range of $T_i$ is [0,1], and the range of $T_d$ is [0,1].

Set the weights $\omega_1 = 0.99$, $\omega_2 = 0.001$, $\omega_3 = 1.9$ through multiple experiments. The optimization process of the performance index fitness function of the objective function and the PID algorithm control step response after parameter tuning and optimization are shown in Figs 15 and 16.

Fig 15 shows that the value of the performance index of the population time error integral function began to decline rapidly around 10–20 generations, indicating that the fitness of the population selected by the genetic algorithm is rapidly improving.

It can be seen from Fig 16 that the traditional PID algorithm has obvious overshoot and its steady-state performance is far inferior to the PID algorithm optimized through genetic algorithm tuning. PID control based on genetic algorithm overcomes some shortcomings of traditional PID. By comparing with the traditional PID algorithm, using the genetic PID algorithm to control the tension can speed up the tension response speed of the system, reduce the overshoot of the system, and achieve the desired effect.

## 6 Experimental test of control system

### 6.1 The test of the embedded soft PLC control system on the tension control of the rapier loom

In the normal weaving process of the QJH910 rapier loom, the warp tension setting value is about 1500N as an example. According to the technological requirements of the fabric to be woven, the upper and lower limits of the allowable tension are generally not more than 10%. Fig 17 is the real-time acquisition interface of the tension curve of the rapier loom on-site debugging.

The commonly used control method in rapier loom warp tension adjustment is to use PID algorithm to adjust it. This paper found through experiments that the use of traditional PID algorithm can ensure that the warp tension error does not exceed 10%. However, the adjustment period is long, the adjustment speed is slow, and the actual tension of the collected warp has relatively large fluctuations. Although there is no moment of exceeding the upper and lower limits of the tension within a certain period of time, the rapier loom may experience excessive or insufficient tension during long-term operation. There are many factors that affect tension, and the use of fixed PID parameters can only avoid unilateral factors. Table 1 is the 10

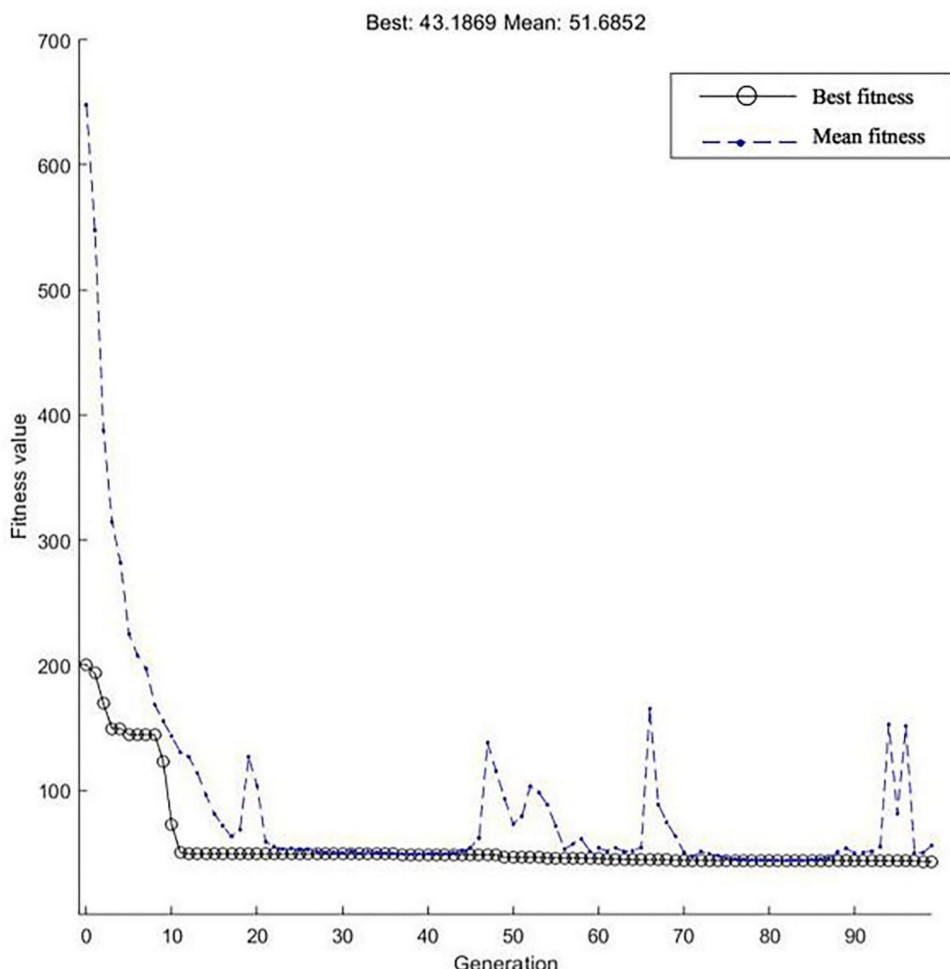

**Fig 15. Genetic algorithms performance index optimization process.**

sets of data collection results of using PID algorithm to adjust the warp tension of the rapier loom.

This paper uses genetic algorithm to find the optimal solution of the problem to optimize the three parameters of PID $K_p$, $T_i$ and $T_d$. According to the difference between the actual warp tension and the set warp tension during the actual operation of the rapier loom, the three parameters $K_p$, $T_i$ and $T_d$ in the PID algorithm are optimized in real time. The principle of the genetic algorithm to optimize PID parameters is that if the measured value of the warp tension of the rapier loom is greater than the set value of the warp tension, the genetic algorithm will determine the $K_p$ according to the difference. The greater the difference, the greater the value of $K_p$, so as to achieve the purpose of rapid adjustment to achieve a stable value. If the difference is relatively small, the value of $K_p$ will be adjusted appropriately, so the system will not cause overshoot due to the value of $K_p$ is too large. Then adjust the values of $T_i$ and $T_d$ according to the error, and adjust the warp tension of the rapier loom in an optimal way. Then through the optimization and adjustment of PID parameters, the PID algorithm can obtain the compensation amount in real time during the process of adjusting the warp tension of the rapier loom. Finally, the warp tension of the rapier loom is controlled

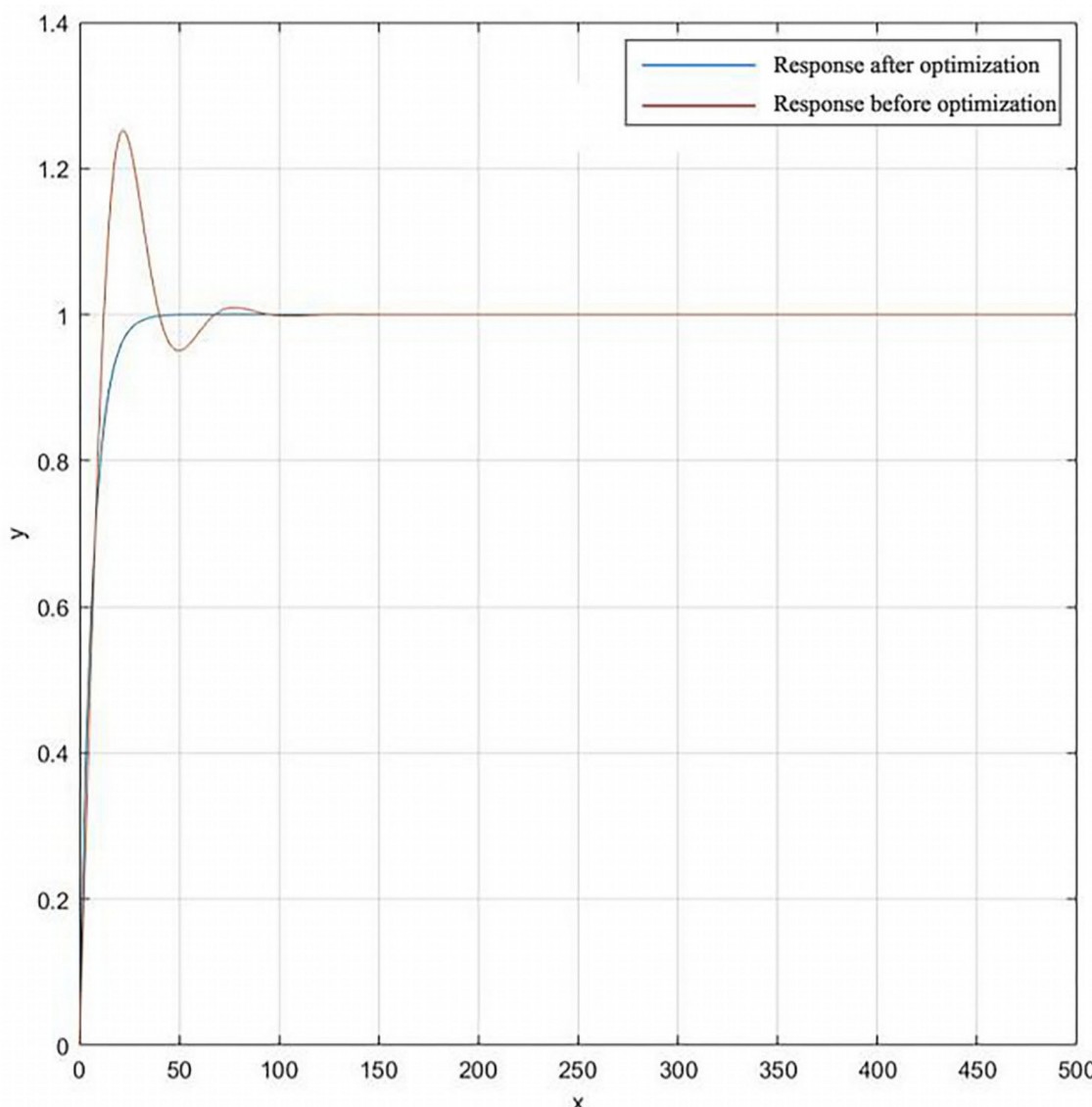

**Fig 16. Step response simulation diagram of PID controller.**

by adjusting the pulse frequency of the let-off servo motor [34–36]. Table 2 is the 10 sets of data collection results of using genetic PID algorithm to adjust the warp tension of the rapier loom.

Through the experiment, 10 randomly selected from 100 experimental data, according to the actual measured value of warp tension under the control of genetic PID algorithm and its error comparison Table 2. We can see that the warp tension error of the rapier loom has been kept within 5% during the operation. And through data analysis, it is proved that the warp tension fluctuation of the rapier loom under the control of the genetic PID algorithm is more stable than that under the control of the traditional PID algorithm, the adjustment period is shorter, and the adjustment speed is faster. Obviously, it is more effective to use genetic algorithm to optimize PID parameters to control the tension of the rapier loom.

**Fig 17. Tension test curve.**

## 6.2 Field system experiment

Through on-site debugging, due to the influence of the QJH910 rapier loom, the spindle speed is 300r/min during debugging. During the weaving process of QJH910 rapier loom, the tension can be maintained at about 150Kg for a long time. The color selection, shedding action, selvedge action, etc. of the weft selector work in accordance with the normal process requirements, and the quality of the woven fabric meets the needs of users.

Moreover, the results of the field operation proved that in the process of using the genetic PID algorithm to control the warp tension of the rapier loom, not only the warp tension fluctuation can be guaranteed to be small, but the error is kept within 5%. And through the

**Table 1. PID algorithm warp tension measured values and error comparison table.**

| Number | Warp tension setting value (Kg) | Warp tension measured value (Kg) | Error (Kg) | Relative error (%) |
|---|---|---|---|---|
| 10 | 150 | 140.76 | -9.24 | 6.16 |
| 20 | 150 | 139.27 | -10.73 | 7.15 |
| 30 | 150 | 156.91 | 6.91 | 4.60 |
| 40 | 150 | 145.70 | -4.30 | 2.87 |
| 50 | 150 | 151.72 | 1.72 | 1.14 |
| 60 | 150 | 163.43 | 13.43 | 8.95 |
| 70 | 150 | 149.19 | -0.81 | 0.54 |
| 80 | 150 | 159.17 | 9.17 | 6.11 |
| 90 | 150 | 160.82 | 10.82 | 7.21 |
| 100 | 150 | 143.52 | -6.48 | 4.32 |

**Table 2. The actual warp tension measured by genetic PID algorithm and its error comparison table.**

| NUMBER | WARP TENSION SETTING VALUE (Kg) | WARP TENSION MEASURED VALUE (Kg) | ERROR (Kg) | RELATIVE ERROR (%) |
|---|---|---|---|---|
| 10 | 150 | 145.21 | -4.79 | 3.19 |
| 20 | 150 | 149.95 | -0.05 | 0.03 |
| 30 | 150 | 155.08 | 5.08 | 3.39 |
| 40 | 150 | 148.51 | -1.49 | 0.99 |
| 50 | 150 | 149.96 | -0.04 | 0.03 |
| 60 | 150 | 151.11 | 1.11 | 0.74 |
| 70 | 150 | 145.53 | -4.47 | 2.98 |
| 80 | 150 | 149.31 | -0.69 | 0.46 |
| 90 | 150 | 147.37 | -2.63 | 1.75 |
| 100 | 150 | 149.86 | -0.14 | 0.09 |

comparison of the products, it is found that the fabric produced under the tension adjustment by the genetic PID algorithm is of higher quality than the fabric produced under the warp tension adjustment by the traditional PID algorithm. It is precisely due to the regulation of the stability of warp yarn tension that the number of times of warp breaks is reduced, the failure rate is reduced, and the weaving efficiency is improved. The debugging site of rapier loom is shown in Fig 18.

## 7 Conclusion

This article mainly describes the research of a rapier loom control system based on embedded soft PLC for the tension control of the rapier loom. Analyze the development status of the rapier loom and its control system and study the development trend content and deficiencies of related research directions. This paper independently developed and designed the embedded control platform of the soft PLC system, and carried out the innovative development and design of the embedded system in terms of improving the programmability of the system, reducing the maintenance cost of the system and further improving the production efficiency of the rapier loom. Then, by comparing the difference between the single-chip control system

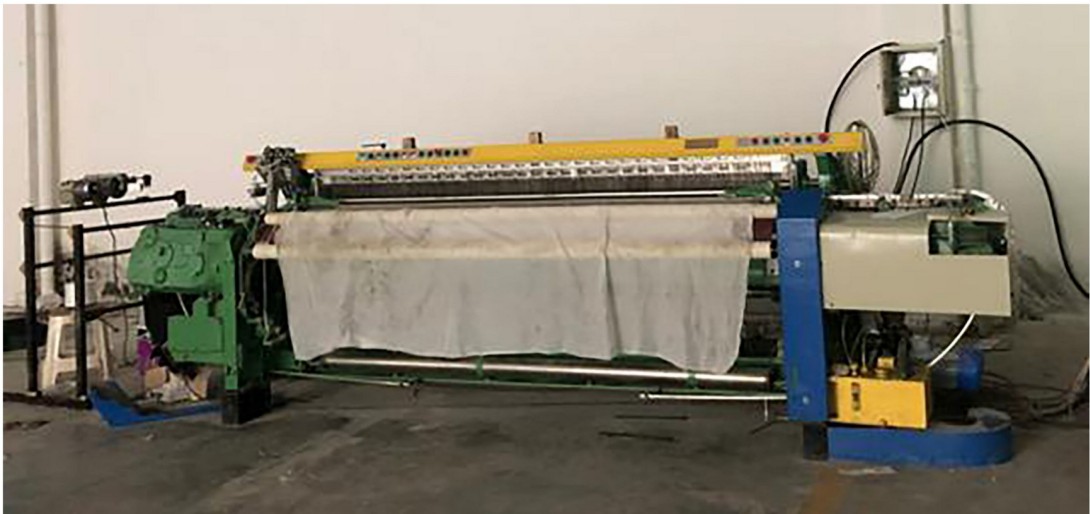

**Fig 18. Debugging site of rapier loom.**

and the PLC control system, it is innovatively proposed to combine the single-chip and PLC to independently develop an embedded soft PLC control system. Finally, this control system is applied to the tension control of the rapier loom.

In this paper, the hardware structure and software structure of the embedded soft PLC control system are designed. At the same time, the embedded soft PLC bottom drive system and the embedded soft PLC control system main program are designed. By using the embedded software PLC hardware structure as the operating platform, the feasibility of the software to control the rapier loom is verified.

In this paper, the analysis of the factors affecting the warp tension of rapier looms shows that the warp tension system is a system with real-time changes, nonlinearity and many interference factors. It is difficult to achieve effective control of warp tension through traditional PID. Therefore, this paper uses genetic algorithm to optimize the PID parameters, realize the effective regulation of the tension system, strengthen the cooperation between the let-off mechanism and the take-up mechanism, and improve the overall stability.

Finally, the system verification of the embedded soft PLC control system from the realization of its functions, the realization of the underlying system and the module of the tension system is done, and the feasibility of the system can be proved by experimental phenomena. The results of experimental verification and on-site debugging prove that the embedded soft PLC control system with programmability and low maintenance cost proposed in this paper can run continuously in the weaving workshop for a long time during the actual control process of the rapier loom. And it can well meet the needs of local enterprises for textile technology.

In the future, the improvement of the embedded soft PLC control system of the rapier loom is mainly reflected in the following aspects:

1. With the advancement of science and technology, in the future development, the research of remote control system is imperative. Therefore, in order to meet future development needs, future embedded soft PLC system designs should add wireless transmission modules or industrial Ethernet modules to meet the needs of industrial interconnection while increasing the transmission rate.

2. The rapier loom system has multiple input and output channels. The design of this system meets the system requirements, but increases the complexity of the system to a certain extent. In the future design of this subject, it is necessary to further develop new solutions to simplify the system structure on the basis of satisfying the system functions and make the whole system more compact.

In order to improve the degree of intelligence of the system, this subject should consider adding the functions of system fault diagnosis and intelligent prediction in the future design process.

## Supporting information

**S1 File.**
(DOCX)

## Author Contributions

**Formal analysis:** Yanjun Xiao, Linhan Shi, Feng Wan, Weiling Liu.

**Methodology:** Yanjun Xiao, Linhan Shi, Wei Zhou.

**Writing – review & editing:** Yanjun Xiao, Linhan Shi.

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
