## [Decision Letter · Decision Letter 0]

3 Jun 2021

PONE-D-21-15942

Application of embedded soft PLC in the control system of rapier loom

PLOS ONE

Dear Dr. Shi,

Thank you for submitting your manuscript to PLOS ONE. After careful consideration, we feel that it has merit but does not fully meet PLOS ONE’s publication criteria as it currently stands. Therefore, we invite you to submit a revised version of the manuscript that addresses the points raised during the review process.

Based on the comments received from the reviewers and my own observation, I recommend major revisions for the article.

We look forward to receiving your revised manuscript.

Kind regards,

Thippa Reddy Gadekallu

Academic Editor

PLOS ONE

Journal Requirements:

4.Thank you for stating the following in the Financial Disclosure section:

"This work was supported by Jiangsu Province training fund funded project (Grant No. BRA2020244)." 

We note that one or more of the authors are employed by a commercial company: Career Leader intelligent control automation company

5. Please ensure that you refer to Figure 1, 2, 4, 6, 8, 9, 10, 11, 12 and 17 in your text as, if accepted, production will need this reference to link the reader to the figure.

Reviewers' comments:

Reviewer's Responses to Questions

**Comments to the Author**

1. Is the manuscript technically sound, and do the data support the conclusions?

Reviewer #1: Yes

Reviewer #2: Yes

2. Has the statistical analysis been performed appropriately and rigorously? 

Reviewer #1: Yes

Reviewer #2: Yes

3. Have the authors made all data underlying the findings in their manuscript fully available?

Reviewer #1: Yes

Reviewer #2: Yes

4. Is the manuscript presented in an intelligible fashion and written in standard English?

Reviewer #1: Yes

Reviewer #2: Yes

5. Review Comments to the Author

Reviewer #1: 1. Introduction section can be extended to add the issues in the context of the existing work

2. Literature review techniques have to be strengthened by including the issues in the current system and how the author proposes to overcome the same.

3. What is the motivation of the proposed work?

4. Research gaps, objectives of the proposed work should be clearly justified.

5. The authors should consider more recent research done in the field of their study (especially in the years 2018 and 2020 onwards). 6. The paper needs to provide significant experimental details to correctly assess its contribution: What is the validation procedure used?

7. Kindly provide several references to substantiate the claim made in the abstract (that is, provide references to other groups who do or have done research in this area).

8. An error and statistical analysis of data should be performed.

9. The conclusion should state scope for future work.

10. Discuss the future plans with respect to the research state of progress and its limitations.

11. Kindly refer the below paper:

1. Rajput, D.S., Basha, S.M., Xin, Q. et al. Providing diagnosis on diabetes using cloud computing environment to the people living in rural areas of India. J Ambient Intell Human Comput (2021). https://doi.org/10.1007/s12652-021-03154-4

Reviewer #2: • In Introduction section, the drawbacks of each conventional technique should be described clearly.

• Introduction needs to explain the main contributions of the work more clearly.

• The authors should emphasize the difference between other methods to clarify the position of this work further.

• The Wide ranges of applications need to be addressed in Introductions

• The objective of the research should be clearly defined in the last paragraph of the introduction section.

• Add the advantages of the proposed system in one quoted line for justifying the proposed approach in the Introduction section. In literature survey under optimization the authors can refer A metaheuristic optimization approach for energy efficiency in the IoT networks. Green communication in IoT networks using a hybrid optimization algorithm

6. PLOS authors have the option to publish the peer review history of their article (what does this mean?). If published, this will include your full peer review and any attached files.

Reviewer #1: No

Reviewer #2: No

---

## [Author Response · Author response to Decision Letter 0]

12 Jul 2021

Dear Editor and Reviewers:

On behalf of my co-authors, we are very grateful to you for giving us an opportunity to revise our manuscript. We appreciate you very much for your positive and constructive comments and suggestions on our manuscript entitled“Application of embedded soft PLC in the control system of rapier loom”(PONE-D-21-15942).

We have studied reviewers’ comments carefully and tried our best to revise our manuscript according to the comments. The following are the responses and revisions. I have made in response to the reviewers’ questions and suggestions on an item-by-item basis. Thank again to the hard work of the editor and reviewer!

Response to the comments of academic editor：

1.The manuscript was changed to meet the style requirements of PLOS ONE.

2.The reference list of the manuscript was checked and ensured that it was complete and correct.

3.The list of authors on the manuscript was revised and ensured that each author had contact with a certain institution.

4.Provided a revised funding statement and a statement about the role of funders in the research. Provides a statement about the authors’ contributions and clearly and accurately indicates the role of these authors in the research.

5.Provides an updated statement of competitive interest. And confirm that this business cooperation will not change compliance with all PLOS ONE policies on sharing data and materials.

6.Make sure to refer to figures 1, 2, 4, 6, 8, 9, 10, 11, 12, and 17 in the text and production will need this reference to link readers to the figures.

Response to the comments of Reviewer #1：

1.Introduction section can be extended to add the issues in the context of the existing work.

Response: In the introduction part of the thesis, the description of the related work of the existing embedded soft PLC system is expanded. And compared the advantages and disadvantages of existing related work and research.

2.Literature review techniques have to be strengthened by including the issues in the current system and how the author proposes to overcome the same.

Response: The literature review in the introduction section adds a comparison of the latest domestic and foreign related research. It highlights the necessity of the embedded soft PLC control system proposed in this paper in reducing the cost of maintenance and improving the programmability of the system. And a simple explanation of how this article solves the problems raised, that is, this article designs the hardware structure and software structure of the embedded soft PLC control system for specific problems. At the same time design the embedded soft PLC bottom drive system and the embedded soft PLC control system main program. Finally, using STM32 single-chip microcomputer as the main control unit and PLC programming method, effectively reducing the maintenance cost of the system later and effectively improving the programmability of the system.

3.What is the motivation of the proposed work?

Response: The proposed motive is to make the rapier loom better realize automated production, further improve the production efficiency of the rapier loom, increase the programmability of the system, and reduce the cost of system maintenance.

4.Research gaps, objectives of the proposed work should be clearly justified.

Response: Through the actual investigation of current rapier loom manufacturers at home and abroad, it is found that there are problems of high later maintenance cost and poor programmability for the rapier loom control system. The comparison of recent domestic and foreign related research shows that the related research literature based on embedded soft PLC has not solved the problems of high maintenance cost and poor system programmability that exist in the control system of rapier looms.

5.The authors should consider more recent research done in the field of their study (especially in the years 2018 and 2020 onwards).

Response: Added the latest domestic and foreign research around 2020 and compared and described its advantages and disadvantages.

6.The paper needs to provide significant experimental details to correctly assess its contribution: What is the validation procedure used?

Response: In the fourth part, the description of the key program analysis process in the embedded soft PLC system mentioned in the thesis is added.

7.Kindly provide several references to substantiate the claim made in the abstract (that is, provide references to other groups who do or have done research in this area)

Response: Added the latest domestic and foreign research around 2020 and compared and described its advantages and disadvantages. In the literature review in the introduction, some comparisons of the latest domestic and foreign related studies are added. It highlights the necessity of the embedded soft PLC control system proposed in this paper in reducing the cost of maintenance and improving the programmability of the system. Finally, this article adds a description of the advantages of the proposed system in the introduction. This article also refers to the two documents "A metaheuristic optimization approach for energy efficiency in the IoT networks" and "Green communication in IoT networks using a hybrid optimization algorithm". In this way, the origin of the design idea of the embedded soft PLC system proposed in this paper is described, and the rationality of the proposed method is proved.

8.An error and statistical analysis of data should be performed.

Response: By adding the traditional PID algorithm to the analysis of the error and the relative error of the warp tension measurement value. Compared with the PID control algorithm based on genetic algorithm adopted in this paper, the error and relative error of the loom control tension measured value are compared. It is found that the use of genetic algorithm to control the rapier loom in the embedded soft PLC control system has significantly reduced errors.This can show that the embedded soft PLC control system proposed in this paper can effectively improve the precision of loom tension control while reducing maintenance costs and improving system programmability, thereby effectively improving the production efficiency of the loom.

9.The conclusion should state scope for future work

Response: In the conclusion, the scope of future work is added to strengthen the research on the remote control system, simplify the system structure, and improve the degree of system intelligence.

10.Discuss the future plans with respect to the research state of progress and its limitations.

Response: This subject is not deep enough in the remote control system, simplifying the system structure, and improving the degree of system intelligence. Therefore, in the future, the improvement of the embedded soft PLC control system of the rapier loom is mainly reflected in the following aspects:

1. With the advancement of science and technology, in the future development, the research of remote control system is imperative. Therefore, in order to meet future development needs, wireless transmission modules or industrial Ethernet modules are added to the design of embedded soft PLC systems. While the transmission rate is increasing, it meets the needs of industrial interconnection. 

2. The rapier loom system has many input and output channels. The design of this system meets the system requirements, but it increases the complexity of the system to a certain extent. In the future design of this subject, it is necessary to further develop new solutions to simplify the system structure on the basis of satisfying the system functions and make the whole system more compact.

 3. In order to improve the degree of intelligence of the system, this subject should consider adding the functions of system fault diagnosis and intelligent prediction in the future design process.

Response to the comments of Reviewer #2： 

• In Introduction section, the drawbacks of each conventional technique should be described clearly.

Response: In the introduction part, a more detailed description of the shortcomings of traditional rapier loom control technology. And through a more detailed comparison and description of the advantages and disadvantages of traditional technologies, it highlights the advantages and innovations of the embedded soft PLC control system studied in this paper.

• Introduction needs to explain the main contributions of the work more clearly.

Response: In the introduction, through describing the advantages and innovations of the embedded soft PLC control system compared with the previous rapier loom control system, the contribution of the research content of this article is highlighted. The contribution and innovation of this article lies in the development of a complete low-cost control system, that is, an embedded soft PLC control system. This system solves the problems of high maintenance cost and poor programmability in the rapier loom control system in the past. After that, based on this system platform, we completed a more effective control of the tension system of the loom through a PID algorithm optimized by genetic algorithm. Finally, we proved the advantages of the system we developed through the on-site debugging results, and verified the rationality and feasibility of the system.

• The authors should emphasize the difference between other methods to clarify the position of this work further

Response: By adding a description of the difference between the method proposed in this article and other methods in the last paragraph of the introduction. That is, compared with other control methods in the field of rapier looms, this system solves the problems of high maintenance cost and poor programmability that have occurred in the control system of rapier looms in the past.

• The Wide ranges of applications need to be addressed in Introductions

Response: A description of the scope of application of the control system proposed in the thesis is added in the last paragraph of the introduction. That is, the application range of the control system proposed in this article is very wide. The control system proposed in this paper is suitable for rapier loom control systems generally used in actual industrial production.

• The objective of the research should be clearly defined in the last paragraph of the introduction section.

Response: In the last paragraph of the introduction, the research goal of this article is pointed out. That is, the research goal of this article is to integrate the advantages of the single-chip control system and the PLC control system, reduce the maintenance cost of the system and improve the programmability of the system, thereby further improving the production efficiency of the rapier loom.

• Add the advantages of the proposed system in one quoted line for justifying the proposed approach in the Introduction section. In literature survey under optimization the authors can refer A metaheuristic optimization approach for energy efficiency in the IoT networks. Green communication in IoT networks using a hybrid optimization algorithm

Response: This article adds a description of the advantages of the proposed system in the introduction and draws reference to the two documents "A metaheuristic optimization approach for energy efficiency in the IoT networks" and "Green communication in IoT networks using a hybrid optimization algorithm". In this way, the origin of the design idea of the embedded soft PLC system proposed in this paper is described, and the rationality of the proposed method is proved.

Changes to the reference list：Added citations and references to the following 4 papers in the article

[13]Xu Xiao. Application of embedded soft PLC in coal mine control system[J]. Energy Technology and Management, 2020, 45(04):181-182.

[14] Gao Yanxiang. Design of embedded soft PLC control system for comprehensive excavation equipment [J]. Coal Mine Machinery, 2020, 41(07): 174-178.

[15]Shi Chunxiao. Intelligent controller design based on embedded soft PLC technology [J]. Computer Measurement and Control, 2020, 28(04): 126-130.

[16] Zhu Wei, Wang Hong, Li Shoubin, Zhao Wensheng. Design of roadheader control system based on embedded soft PLC [J]. Industry and Mine Automation, 2020, 46(02): 100-106.

We look forward to hearing from you regarding our submission. We would be glad to respond to any further questions and comments that you may have.

---

## [Decision Letter · Decision Letter 1]

6 Sep 2021

Application of embedded soft PLC in the control system of rapier loom

PONE-D-21-15942R1

Dear Dr. Shi,

We’re pleased to inform you that your manuscript has been judged scientifically suitable for publication and will be formally accepted for publication once it meets all outstanding technical requirements.

Kind regards,

Yogendra Arya

Academic Editor

PLOS ONE

Additional Editor Comments (optional):

Reviewers' comments:

Reviewer's Responses to Questions

**Comments to the Author**

1. If the authors have adequately addressed your comments raised in a previous round of review and you feel that this manuscript is now acceptable for publication, you may indicate that here to bypass the “Comments to the Author” section, enter your conflict of interest statement in the “Confidential to Editor” section, and submit your "Accept" recommendation.

Reviewer #1: All comments have been addressed

2. Is the manuscript technically sound, and do the data support the conclusions?

Reviewer #1: Yes

3. Has the statistical analysis been performed appropriately and rigorously? 

Reviewer #1: Yes

4. Have the authors made all data underlying the findings in their manuscript fully available?

Reviewer #1: Yes

5. Is the manuscript presented in an intelligible fashion and written in standard English?

Reviewer #1: Yes

6. Review Comments to the Author

Reviewer #1: 1. The study presents the results of original research.

2. Results reported have not been published elsewhere.

3. Experiments, statistics, and other analyses are performed to a high technical standard and are described in sufficient detail.

4. Conclusions are presented in an appropriate fashion and are supported by the data.

5. The article is presented in an intelligible fashion and is written in standard English.

6. The research meets all applicable standards for the ethics of experimentation and research integrity.

7. The article adheres to appropriate reporting guidelines and community standards for data availability.

7. PLOS authors have the option to publish the peer review history of their article (what does this mean?). If published, this will include your full peer review and any attached files.

Reviewer #1: No

---

## [Editor Report · Acceptance letter]

10 Sep 2021

PONE-D-21-15942R1 

Application of embedded soft PLC in the control system of rapier loom 

Dear Dr. Shi:

I'm pleased to inform you that your manuscript has been deemed suitable for publication in PLOS ONE. Congratulations! Your manuscript is now with our production department. 

Kind regards, 

on behalf of

Dr. Yogendra Arya 

Academic Editor

PLOS ONE